# Efficacy of a High-Iron Dietary Intervention in Women with Celiac Disease and Iron Deficiency without Anemia: A Clinical Trial

**DOI:** 10.3390/nu12072122

**Published:** 2020-07-17

**Authors:** Alice Scricciolo, Luca Elli, Luisa Doneda, Karla A Bascunan, Federica Branchi, Francesca Ferretti, Maurizio Vecchi, Leda Roncoroni

**Affiliations:** 1Center for Prevention and Diagnosis of Coeliac Disease-Gastroenterology and Endoscopy Unit, Fondazione IRCCS Ca’ Granda Ospedale Maggiore Policlinico, Via F. Sforza 35, 20122 Milan, Italy; lucelli@yahoo.com (L.E.); federica.branchi@gmail.com (F.B.); francesca.ferretti01@gmail.com (F.F.); maurizio.vecchi@unimi.it (M.V.); leda.roncoroni@unimi.it (L.R.); 2Department of Pathophisiology and Transplantation, University of Milan, 20122 Milan, Italy; 3Department of Biomedical, Surgical and Dental Sciences, University of Milan, 20122 Milan, Italy; luisa.doneda@unimi.it; 4Department of Nutrition, School of Medicine, University of Chile, 8380453 Santiago, Chile; karlabascunan@gmail.com

**Keywords:** Celiac disease, iron deficiency without anemia, dietary iron, iron supplementation, gluten-free diet, women

## Abstract

Background and Aim. Iron deficiency without anemia (IDWA) is a common finding in celiac disease (CD) and can also persist in case of good compliance and clinical response to a strict gluten-free diet (GFD). This scenario usually presents in CD women of child-bearing age in whom the imbalance between menstrual iron loss and inadequate iron intake from their diet plays the major role. A recommended approach to this condition is yet to be established. This study aimed to compare, in this subset of patients, the efficacy of a dietary approach consisting of an iron-rich diet against the traditional pharmacological oral-replacement therapy. Material and Methods. Between February and December 2016, consecutive CD female patients of child-bearing age as referred to our outpatient center with evidence of IDWA (ferritin <15 ng/mL or 15–20 ng/L with transferrin saturation <15%) were enrolled. After the completion of a 7-day weighed food intake recording to assess the usual iron dietary intake, the patients were randomized in two arms to receive a 12-week iron-rich diet (iron intake >20 mg/die) versus oral iron supplementation with ferrous sulfate (FS) (105 mg/day). Blood tests and dietary assessments were repeated at the end of treatment. The degree of compliance and tolerability to the treatments were assessed every month by means of specific questionnaires and symptoms evaluation. Results. A total of 22 women were enrolled and divided in the diet group (*n* = 10, age 37 ± 8 years) and in the FS group (*n* = 12, age 38 ± 10 years). The food intake records demonstrated an inadequate daily intake of iron in all the enrolled subjects. At the end of the treatments, ferritin levels were higher in the FS group (8.5 (5) versus 34 (30.8), *p* = 0.002). Compliance and tolerability were similar in both treatment groups (89% versus 87%, *p* = ns). Conclusions. These findings did not support any equivalent efficacy of an iron-rich diet compared to a FS supplementation in non-anemic iron-deficient women affected by CD. However, the diet appeared a well-tolerated approach, and adequate dietary instructions could effectively increase the daily iron consumption, suggesting a role in the long-term management of IDWA, especially in patients who do not tolerate pharmacological supplementation.

## 1. Introduction

Celiac disease (CD) is an autoimmune disorder that occurs in genetically predisposed individuals who develop an immune reaction to gluten [1]. In the Western countries, the prevalence has reached 1:100, with a male/female ratio 1/3 [2,3]. The CD hallmark consists in a damage of the gastrointestinal tract characterized by inflammation of the lamina propria, villous atrophy, crypt hyperplasia, and T-cell infiltration [4]. The clinical manifestations of CD are heterogeneous. The classic ones involve gastrointestinal-related symptoms due to malabsorption, mainly diarrhea and weight loss, but up to 30% of patients are asymptomatic. Furthermore, CD patients may present extra-intestinal symptoms, including mineral and vitamin deficiency. The most common mineral deficiencies are: iron-deficiency anemia (IDA) and iron-deficiency without anemia (IDWA) [5,6,7,8].

IDA is the most frequently sign observed in patients affected by CD, and different studies have shown that IDA can be the only symptom of CD [8,9,10]. Although not completely clear, iron deficiency (ID) in CD can be caused by malabsorption due to small bowel (SB) atrophy, which is a systemic inflammatory state and possible genetic variants. While most of the studies focalized on IDA, little is known about IDWA. Gluten-free diet (GFD) is able to restore intestinal trophism in the majority of patients, re-establishing the correct absorption of dietary iron [11] and reducing the cytokine levels. However, in spite of the intestinal mucosa normalization and clinical responsiveness, in a group of CD patients, IDWA may persist. Frequently, these patients are women of child-bearing age with menstrual blood losses and a subsequently progressive depletion of iron reserves. In the case of IDA, the current recommendations are to take pharmacological iron supplementation, in order to prevent the depletion of the iron deposits. It has been observed that the normalization of hemoglobin levels requires up to two years of therapy [12]. Experts do not recommend any specific treatment to IDWA patients, and there are no guidelines on this issue; consequently, it is unclear if this group of patients should be pharmacologically treated as in the case of IDA. Providing high doses of iron could not be a correct strategy in this group; in fact, iron overload may induce such negative effects as the formation of free radicals and protein and DNA damages [13,14]. Conversely, to increase the iron dietary intake changes in gluten-free diet (GFD) can restore iron deposits. From this point of view, the daily amount of iron recommended by the World Health Organization (WHO) varies according to the age and sex of individuals, being around 20 mg per day for women [15].

The aim of our clinical trial was to evaluate the efficacy of a high-iron gluten-free diet (GFD-HI) as compared to an oral iron therapy (ferrous sulfate) in a group of women affected by CD with IDWA.

## 2. Patients and Design

Our study patients were recruited at the “Center for Prevention and Diagnosis of Celiac Disease”, Fondazione IRCCS Ca’ Granda Ospedale Maggiore Policlinico in Milan between 1 February and 31 December 2016. CD diagnosis was based in accordance with the national and international guidelines [8]. Celiac adult women of pre-menopausal age, following a correct GFD for at least 1 year, presenting with IDWA (defined as ferritin levels <15 ng/L or ferritin 15–20 ng/L and transferrin saturation <15%), were enrolled.

The study was a prospective two-arm single-center randomized open-label trial. At enrolment (t0), the patients’ demographic and clinical data were recorded; furthermore, a 7-day food diary was kept in order to verify the iron dietary intake. Blood tests including Hb, ferritin, iron, transferrin, anti tissue transglutaminase antibody were performed, and gastrointestinal symptoms were evaluated by means of 10-cm-long visual analogic scales (VAS). After these evaluations, the subjects were divided in two treatment arms for 12 weeks: (1) patients receiving a GFD with high-iron content, >20 mg per day; (2) patients receiving ferrous sulfate (FS) (105 mg/day, 1 oral tablet). The subjects were assigned to either treatment group on the basis of a randomization list generated by computer. All the patients were evaluated by a gastroenterologist and a qualified nutritionist: both experts in CD management. At the end of the treatment (t3), the patient’s blood tests, gastrointestinal symptoms (VAS), and 7-day food diary were acquired. After 4 and 8 weeks from the beginning of their treatment, the patients were contacted by telephone in order to ascertain their adherence to the treatments and symptoms. The study flowchart has been reported in Figure 1.

The study was registered on http://clinicaltrials.gov/ (ref. no. NCT02949765). The University of Milan’s Institutional Review Board checked and approved the study protocol according to the Helsinki Declaration, the Project Identification Code of the Local Ethics Committee’s Approval of our study being 744_2015bis. The protocol was approved by the Ethics Committee of Milan/Area B on 14 January 2016). All the patients gave and signed their informed consent prior to participation in this study.

### High-Iron Gluten-Free Diet

The subjects underwent an interview with a qualified nutritionist about the content and bio-availability of iron in the different foods. Foods were divided into three categories depending on their iron content (high, medium, and low), as identified by a previous study [16]. The quantity of iron contained in each food was determined via the Italian food composition tables [17]. The types of gluten-free (GF) foods in each category are shown in Appendix A. A nutritionally balanced GFD was designed with a combination of animal and vegetable food sources. The patients were advised to eat meals with a high intake of vitamin C to increase the absorption of iron, to limit fiber and avoid coffee, tea, or milk near mealtimes, in order to preserve a regular iron absorption [18,19,20]. To choose a diet with high-iron content, the patients were recommended to select one of the following four daily food combinations, with a specific number of portions per category: 1 high + 2 medium + 2 low, 1 high + 1 medium + 4 low, 4 medium + 2 low, 3 medium + 4 low. Each combination of foods ensured an intake of at least 20 mg/day iron. To verify the effective compliance with the assigned diet, the patients received questionnaires to be completed 15 times over the study period, about the daily food combination and about the advices to increase the absorption of iron.

## 3. Statistical Analysis 

The data were described as median (interquartile range). The data distribution was assessed by graphical inspection and the Shapiro–Wilk test. Wilcoxon matched-pairs signed-ranks test used to compare iron status and gastrointestinal symptoms were reported by the patients within the groups before and after intervention. A Wilcoxon rank-sum test was used to compare gastrointestinal symptoms at the end of the intervention between groups. The VAS score for gastrointestinal symptoms before and after the intervention were analyzed by Analysis of Variance (ANOVA, including factors ‘group’ and ‘time’) for repeated measures (‘time’). A 5% significance level was used, and the software packages STATA^®^ v. 13.1 (StataCorp LLC, College Station, TX, USA) and GraphPad Prism v. 6 (GraphPad Software, La Jolla, CA, USA) were used for analysis and graphs processing.

## 4. Results

Twenty-two CD women with IDWA were enrolled and allocated to the GFD-HI group (*n* = 10, age 37 ± 8, mean age at diagnosis 27.1 ± 11.5) and to the FS group (*n* = 12, age 38 ± 10, mean age at diagnosis 29.3 ± 16). At enrolment, most of the patients (77.3%) presented with an insufficient iron dietary intake (7.37 ± 2.27 mg/day).

Hematological data at enrolment and at the end of treatment are reported in Table 1. It is appreciated that the pharmacological treatment significantly improved all blood parameters regarding the iron status. However, in the case of the GFD-HI group, the values at the end of intervention did not show an increase in iron indicators, showing only a tendency to improve ferritin levels in the women following a GFD-HI.

Regarding gastrointestinal symptoms, there were no significant differences when comparing the start and end of treatment in both groups (Table 2). However, it should be noted that for the FS group, there was a statistical tendency to high frequency of diarrhea around the start of the intervention.

Compliance and tolerability were similar in both treatments, with no patients suspending or interrupting the study (data not shown). The VAS scores regarding the symptoms reported by the patients during the trial are shown in Table 2 and Appendix A.

## 5. Discussion

The present study is the first one evaluating the effectiveness of a GFD-HI in patients affected by CD and IDWA. The results of the study have shown that prescribing a diet with high iron content, although not sufficient to normalize the ferritin values in a group of CD women of child-bearing age, is still able to stabilize the levels of Hb, ferritin, and transferrin saturation, while previous studies on other groups of patients reported positive results, suggesting a non-inferiority of high-iron dieting compared to FS administration, especially on long-term observation [16]. On the other side, FS is a validated treatment with proven efficacy on iron deficiency, and the daily food intake is about 1/10^th^ of the quantity present in the drug. Furthermore, in our study, the FS supplementation has been well tolerated, and the analysis of symptoms demonstrated only a slight, non-significant decrease of VAS in the GFD-HI group as compared to the FS group.

While the international guidelines mainly deal with IDA, the diagnostic and therapeutic approach it is not clear in case of IDWA [21]. Treating this condition might play a preventive role, especially in women of child-bearing age, as such women are exposed to the risk of developing IDA, because of the menstrual blood loss. However, oral iron supplementation can cause side effects and lead to the development of symptoms. In this study, the prescription of a GFD-HI involved a group of young women affected by CD. Generally, CD patients are more exposed to nutrients deficiency even after a correct GFD. Their iron deficiency in CD can be attributed to malabsorption, resulting from the persistence of villous atrophy, dietary mistakes, or the presence of genetic polymorphisms that limit iron absorption [22]. In addition, the use of GF industrially manufactured products has been reported to limit the intake of micronutrients as compared to the diet of the general population [23,24]. Iron is necessary for the proper functioning of different cellular mechanisms, including enzymatic processes, DNA synthesis, and mitochondrial energy production. The body of an adult contains 3–5 g iron, 20–25 mg are needed on a daily basis for the production of red blood cells and for cellular metabolism [25]. About 1–2 mg iron is lost daily because of epithelial desquamation, sweating, urinary secretion, and menstrual flow in women. Losses are usually balanced by the intestinal absorption of iron taken through diet and metabolic recycling [26]. The daily intake in milligrams of iron recommended by the World Health Organization (WHO) varies according to the age and sex of individuals, being ca. 20 mg per day for women [15]. A diet with high iron favors foods of animal source such as meat, offal, fish, seafood [27]. It was also advised to take such vegetable-based foods as legumes (beans, chickpeas, lentils), dried fruits (pistachios, cashew nuts, peanuts, dried apricots) and such other foods as rocket, dark chocolate, buckwheat, and olives, although their iron bio-availability is less than foods of animal source [28]. Furthermore, it was recommended to take fruit and vegetables rich in vitamin C such as kiwi, strawberries, citrus fruit, currants, papaya, red and yellow peppers, broccoli, cauliflower, and parsley at mealtimes, thanks to the potential of the ascorbic acid therein contained to favor the absorption of iron [18]. On the contrary, it was not recommended to take coffee, tea, or milk near mealtime, as their content in phenolic compounds (e.g., phytates, oxalates) and calcium may reduce iron absorption [19,29]. It was also advised to limit fiber intake, as fiber interferes with the regular absorption of iron [20]. On analyzing the 7-day food diaries of the enrolled patients, it showed that most patients did not present a sufficient iron intake through their diets.

In case of IDWA, the appropriate intervention is to ensure an adequate daily intake of iron that is sufficient to compensate its consumption. The choice of a therapeutic diet as an alternative to oral iron supplementation appears to be a potentially adequate option for patients with CD. In fact, they often have gastrointestinal symptoms [30], so the prescription of drugs with potential gastrointestinal effects (such as FS, currently considered as the standard therapy for ID and IDA) may be poorly tolerated or even contraindicated.

To achieve significant improvements with the diet in serum ferritin requires time (over nine months), so future studies need longer investigate times. In conclusion, the results of our pilot randomized trial on the efficacy of the GFD-HI in women of child-bearing age with CD do not currently allow for comparable efficacy between a high-iron diet over 12 weeks and the reference standard, i.e., the FS therapy. However, the diet was well tolerated and accepted by celiac patients, maintaining stable iron parameters and blocking any decreasing tendency. For this group of IDWA patients, it would be helpful to test a “run-in period” of 3 weeks of ferrous sulfate supplementation to stabilize their iron profiles and to recommend a high-iron diet that can be followed in the long-term, maintaining the iron blood values as stable (see Figure 2). Further data is expected from the long-term observation of the groups of patients enrolled, in order to highlight an effect on dietary habits, on the effective daily food intake of iron and on the effect of these changes on the iron profile. The support of expert nutritionists in the management of patients with CD appears indispensable not only to ensure an optimal approach to GFD, but also to identify early signs of malnutrition or malabsorption and to provide specific support in the adoption of correct eating habits toward the prevention of micronutrient deficiency, to which celiac subjects are particularly predisposed.

## Figures and Tables

**Figure 1 nutrients-12-02122-f001:**
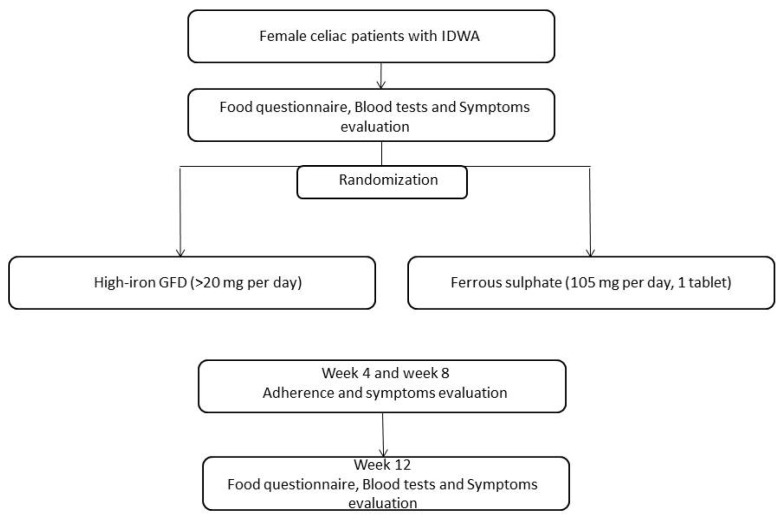
Flowchart of the study (*n* = 22).

**Figure 2 nutrients-12-02122-f002:**
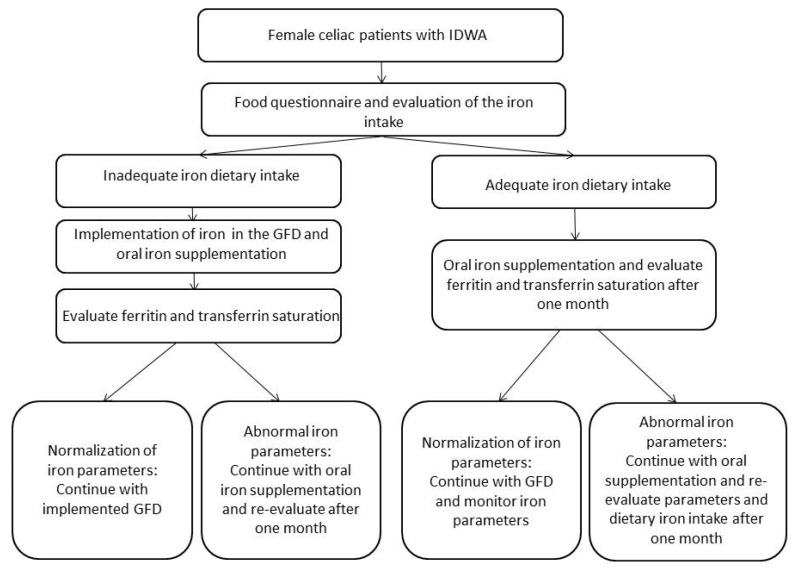
Roadmap in case of iron deficiency without anemia (IDWA) and celiac disease (CD).

**Table 1 nutrients-12-02122-t001:** Blood tests of the enrolled patients.

GFD-HI Group (*n* = 10)			FS Group (*n* = 12)			
	t0	t3	*p*	t0	t3	*p*
Ferritin (ng/mL)	9 (4)	9 (5.2)	0.26	8.5 (5)	34 (30.8)	0.002
Hemoglobin (g/dL)	12.9 (0.4)	12.9 (1.2)	0.72	12.9 (0.6)	13.8 (1.0)	0.03
Iron (mcg/dL)	59 (53)	61 (58)	0.46	51 (37)	98 (27.5)	0.03
Transferrin (mg/dL)	314 (51)	300 (72)	0.06	304 (75.5)	256.5 (32.5)	0.002
Transferrin saturation (%)	14 (6)	10 (13)	0.14	12 (9.5)	24.5 (11)	0.007

Data described as median (interquartile range). Wilcoxon matched-pairs signed-ranks test was used to compare iron status within the groups before and after intervention. t0: enrolment, t3: 12 weeks, GFD-HI group: gluten-free diet with high-iron content group, FS group: ferrous sulfate group.

**Table 2 nutrients-12-02122-t002:** Gastrointestinal symptoms reported by the patients during the treatments.

GFD-HI Group (*n* = 10)				FS Group (*n* = 12)			
	t0	t3	*P* *	t0	t3	*P* *	*P* †GFD-HI vs. FS at t3
Abdominal pain	0 (3)	0 (1)	0.85	0 (1.5)	0.5 (2)	0.35	0.581
Epigastric burning	0 (-)	0 (-)	-	0.5 (4)	0 (2.5)	0.58	0.056
Abdominal bloating	4 (5)	1 (4)	0.22	2.5 (5.5)	0.5 (4)	0.10	0.964
Diarrhea	0 (0)	0 (0)	0.31	0 (0)	0 (2)	0.04	0.300
Constipation	0 (0)	1 (5)	0.28	0 (4)	0 (5)	1.0	0.547

* Data described as median (interquartile range). Wilcoxon matched-pairs signed-ranks test used to compare gastrointestinal symptoms reported by the patients within the groups before and after intervention. † Wilcoxon rank sum test was used to compared gastrointestinal symptoms at the end of the intervention between group. t0: enrolment, t3: 12 weeks, GFD-HI group: gluten-free diet with high-iron content group, FS group: ferrous sulfate group.

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
