# Peer review of "Efficacy of a High-Iron Dietary Intervention in Women with Celiac Disease and Iron Deficiency without Anemia: A Clinical Trial"

_nutrients, 2020, doi:10.3390/nu12072122_

Round 1

Reviewer 1 Report

The abbreviations need to be re-checked, e.g. "Coeliac disease is an autoimmune disorder " as well as the punctuation marks "the only symptom of CD [8, 9, 10]; Although not completely clear".

For me it's not clear if the food questionnaire included items about nutrients like calcium (milk) or vitamin C (citrus) that might influence the iron absorption.

I don't think that the Supplementary Table 1 is needed for the present article, it is not related to the study results and the needs in children or adolescents is not important here.

Some of the variables presented don't seem to have a normal distribution (e.g. "Iron (mcg/dL) 77.37 ± 46.05" or "Constipation 0.77 ± 0.77 2.42 ± 1.04"), so these should be presented as median and interval and appropriate test (Mann-Whitney) be used to compare between 2 groups.

A limitations paragraph is needed.

Author Response

Point 1: The abbreviations need to be re-checked, e.g. "Coeliac disease is an autoimmune disorder " as well as the punctuation marks "the only symptom of CD [8, 9, 10]; Although not completely clear".

Response 1: Abbreviations have been checked, as well as the punctuation marks. Changes have been made in the text.

Point 2: For me it's not clear if the food questionnaire included items about nutrients like calcium (milk) or vitamin C (citrus) that might influence the iron absorption.

Response 2: The food questionnaire included items about nutrients which increase or limit iron absorption. In the text at line 108-109, has been explained better.

Point 3: I don't think that the Supplementary Table 1 is needed for the present article, it is not related to the study results and the needs in children or adolescents is not important here.

Response 3: Thank you for your suggestion, I have proceeded to remove the Supplementary Table 1.

Point 4: Some of the variables presented don't seem to have a normal distribution (e.g. "Iron (mcg/dL) 77.37 ± 46.05" or "Constipation 0.77 ± 0.77 2.42 ± 1.04"), so these should be presented as median and interval and appropriate test (Mann-Whitney) be used to compare between 2 groups.

Response 4: Thank you for this comment. As suggested, the data presented in Tables 1 and 2 have been re-analyzed using non-parametric statistical tests. Data are now presented as median and interquartile range. Likewise, this change was described in the statistical analysis section.

Point 5: A limitations paragraph is needed.

Response 5: A limitations paragraph has been added in the text at line 204-205.

Reviewer 2 Report

Thank you for the opportunity to review this study.

I recommend the authors include an overview of the dietary pattern/model of participants as this can be significant in terms of iron intake - were participants omniverous, or following vegetarian or vegan diets? or was this information not obtained  - say so if not.

I recommend the authors make it clear how they categorised the level of iron in food (line  99 outlines 3 categories), was this high medium and low or do with iron type haem versus non haem iron sources etc. It mentions 3 categories but I am unclear what they are. Can the also clarify if vitamin C increases absorption of all iron in food or just non haem and if the recommendation for participants to have vitamin C reflected this?

I recommend that the phenolic compounds affecting iron absorption are named (phytates, oxalates). Antagonistic effect of calcium might also be mentioned.

I have a few minor comments on English Language:

  • Line 19 diet - reads better than dieting?
  • Line 47 mineral/vitamin deficiency reads better than minerals/vitamins 
  • Line 101 - vegetable not vegetal
  • Line 153 stabilise not stable

Author Response

Point 1: I recommend the authors include an overview of the dietary pattern/model of participants as this can be significant in terms of iron intake - were participants omniverous, or following vegetarian or vegan diets? or was this information not obtained  - say so if not.

Response 1: All of participants were omnivorous. During the enrollment phase was investigated if each patient was omnivorous or vegetarian/vegan. This information was checked analyzing the food questionnaire.

Point 2: I recommend the authors make it clear how they categorised the level of iron in food (line  99 outlines 3 categories), was this high medium and low or do with iron type haem versus non haem iron sources etc. It mentions 3 categories but I am unclear what they are. Can the also clarify if vitamin C increases absorption of all iron in food or just non haem and if the recommendation for participants to have vitamin C reflected this?

Response 2: The level of iron in food was identified from the Italian food composition table, studied by Crea Society [CREA. Centro di ricerca Alimenti e Nutrizione - Tabelle di composizione degli alimenti  https://www.crea.gov.it/alimenti-e-nutrizione. Available online: https://www.crea.gov.it/alimenti-e-nutrizione]. Food classification was based on iron content (high, medium, low) from previous studies (Patterson, A.J.; Brown, W.J.; Roberts, D.C.; Seldon, M.R. Dietary treatment of iron deficiency in women of childbearing age. Am J Clin Nutr 2001, 74, 650-656).

The three categories were high, medium and low content of iron per portion of food (and not per 100 g of food). We checked from the food questionnaire the compliance at the diet and to the advices, including the consume of food with vitamin C content during mealtime. The compliance of all patients was high (89%), so is it possible to speculate that vitamin C had increased absorption of all iron in food.

Text has been modified at line 99-100.

Point 3: I recommend that the phenolic compounds affecting iron absorption are named (phytates, oxalates). Antagonistic effect of calcium might also be mentioned.

Response 3: Text has been modified at line 195-196 and has been added a new reference about the antagonistic effect of calcium.

Point 4: I have a few minor comments on English Language:

  • Line 19 diet - reads better than dieting?
  • Line 47 mineral/vitamin deficiency reads better than minerals/vitamins 
  • Line 101 - vegetable not vegetal
  • Line 153 stabilise not stable

Response 4: Changes have been made in the text, at the correspondent lines.

Reviewer 3 Report

The authors have submitted the manuscript on the evaluation of pharmacological iron supplementation vs consumption of iron-rich food asa treatment for IDWA in Celiac disease patients on GFD. The authors have explained the study, its goals and the design, methodology and outcome has been well written.

Minor comments:

The number of patients in this pilot-study is low, and blood test parameters (Table 1) have very high Standard deviations and hence may bias the interpretation.

The authors could show the blood test parameters as a scatter-plot for better clarity.

Although the authors have stated that the patients were on a GFD and randomly assigned to the two treatment groups. It may be possible that some of the patients may not have regained their normal villi. I would suggest the authors to also include the Marsh scores for individual patients.

Author Response

Point 1: The number of patients in this pilot-study is low, and blood test parameters (Table 1) have very high Standard deviations and hence may bias the interpretation.

Response 1: We agree with this issue and we have now re-analyzed hematological data (table 1) using non-parametric statistical tests. Data are now presented as median and interquartile range.

Point 2: The authors could show the blood test parameters as a scatter-plot for better clarity.

Response 2: The data shown in Tables 1 and 2 were reviewed and modified according to the Reviewers' comments. We believe that the data now shown as median and range in both tables are clearer and more appropriate.

Point 3: Although the authors have stated that the patients were on a GFD and randomly assigned to the two treatment groups. It may be possible that some of the patients may not have regained their normal villi. I would suggest the authors to also include the Marsh scores for individual patients.

Response 3: A requirement of the study was that patients had been on GFD for at least one year, to define clinical responsiveness. Actual guideline do not recommend a routine use of follow up duodenal biopsy neither in the IDWA scenario; consequently, in our study we did not perform a follow up biopsy. For your knowledge, we checked out from our databases and we found data regarding 7 patients, with no sign of atrophic lesions.

Round 2

Reviewer 1 Report

Nothing to comment.